# Microwave Sensitivity Enhanced Asphalt Mastic with Magnetite Powder and Its Performance after Microwave Heating

**Weixiao Yu, Letao Zhang, Yinghao Miao *, Zhenlong Gong and Sudi Wang**

National Center for Materials Service Safety, University of Science and Technology Beijing, Beijing 100083, China; yuweixiao@xs.ustb.edu.cn (W.Y.); zhangletao@xs.ustb.edu.cn (L.Z.); gongzl@xs.ustb.edu.cn (Z.G.); sudiwang@xs.ustb.edu.cn (S.W.)
* Correspondence: miaoyinghao@ustb.edu.cn

**Abstract:** Microwave heating technology is a promising method for asphalt pavement maintenance and de-icing; however, it requires the material to have a good microwave-absorbing ability and can also result in asphalt aging. It is therefore important to develop microwave-sensitive materials used for asphalt pavement maintenance and study the effects of microwave heating on asphalt aging. This study evaluates the electromagnetic characteristics of limestone powder and magnetite powder and explores the influence of microwave heating on the high-temperature rheological and fatigue properties of microwave sensitivity enhanced asphalt mastic with magnetite powder. A vector network analyzer was used to measure the electromagnetic characteristics of limestone powder and magnetite powder. The magnetite filler asphalt mastics were prepared and subjected to microwave heating for 1 h, 2 h, 3 h, and 4 h. Temperature sweep tests, frequency sweep tests, and linear amplitude sweep (LAS) tests were conducted for magnetite filler asphalt mastics before and after microwave heating. LAS experimental results were analyzed based on viscoelastic continuum damage (VECD) theory. The results show that magnetite powders have better electric field energy storage ability, higher dielectric loss and magnetic loss, and better microwave heating efficiency. The complex shear modulus (G*) and rutting factor (G* $\times$ (sin $\delta$)$^{-1}$) rapidly decrease with the increase in temperature, indicating that the mastics' ability to resist deformation decreases sharply. The longer the microwave heating time for magnetite filler asphalt mastics, the faster the high-temperature rheological properties decreased as the temperature rose. The fatigue life of magnetite filler asphalt mastics significantly decreases with the increase in strain and microwave heating time. It is suggested to add anti-aging agents into asphalt materials to reduce the aging effect in the process of microwave heating. This study provides a reference for the application of microwave heating technology in asphalt pavement maintenance.

**Keywords:** asphalt aging; electromagnetic characteristic; fatigue property; high temperature rheological property; magnetite filler asphalt mastic; microwave heating

## 1. Introduction

Asphalt pavements are comfortable, smooth, produce little noise, and are the primary form of road pavement. With the influence of climate and vehicle loads, asphalt pavements will inevitably suffer various forms of deterioration, such as cracks, potholes, and ruts. Freezing rain and snowy weather easily lead to icy roads, presenting potential safety hazards to traffic. Traditional pavement maintenance and de-icing methods are disadvantageous due to their inefficiency and serious environmental pollution [1–3]. Microwave heating technology has the advantages of rapidity, uniformity, and low pollution [4,5], and its application in asphalt pavements has received greater attention recently. However, it requires that the material has a good microwave-absorbing ability and can also result in asphalt aging. Therefore, it is necessary to explore microwave-sensitive materials

used for asphalt pavement maintenance and study the effects of microwave heating on asphalt aging.

Microwaves are an electromagnetic wave with a frequency of 300 MHz to 300 GHz and a wavelength of 1 mm to 1 m. Microwave heating mechanisms include polarized relaxation loss and conductive loss of materials, and thermal energy is generated by the energy conversion within the electromagnetic field [6]. The microwave sensitivity of materials is closely related to their electromagnetic characteristics and determines the microwave heating efficiency [7]. Furthermore, the matching attenuation features of materials are also important for achieving efficient microwave absorption [8]. Traditional asphalt mixtures have poor electromagnetic wave absorption abilities [9], which limits the application of microwave heating technology in asphalt pavement. Researchers have developed and added various microwave-sensitive additives into asphalt mixtures to improve their microwave heating efficiency, such as silicon carbide [10], activated carbon powder [11], manganese dioxide powder [12], steel wool fibers [13], steel slag [14,15], magnetite powders [16], and so on. Magnetite powders, with their high dielectric properties, low costs, and high abundance of material, are promising, microwave-absorbing materials and have been widely used in practice [17–19].

Researchers have achieved abundant results in studying asphalt pavement self-healing (APSH) and de-icing using microwave heating, including the effect of microwave-sensitive additives on APSH [10,20], the evaluation indicators of APSH [21], the effect of microwave heating modes and time on APSH [22–24], microwave de-icing mechanisms [25], and microwave de-icing characteristics [26]. Asphalt aging due to microwave heating has also attracted researchers' attention. Lou et al. [27] and Fernández et al. [28] found that the physical and rheological properties of asphalt binders cause regression after microwave heating treatment. Microwave radiation also leads to the changing of sulfoxide groups. Sha et al. [29] found that microwave radiation causes a distinct aging impact on binders during the first 10 cycles, after which the values become constant. Flores et al. [30] compared asphalt aging induced by microwave heating to infrared radiant heating through tests of penetration, softening points, and rheological properties. Their results showed that microwave heating has less of an influence on asphalt aging than infrared radiant heating. However, the studies on asphalt aging caused by microwave heating are still insufficient. Little attention has been paid to the high-temperature rheological and fatigue properties of asphalt mastics with magnetite powder after microwave heating, for example.

The dynamic shear rheometer (DSR) is usually used to analyze the high temperature rheological and fatigue properties of asphalt binders and asphalt mastics using temperature sweep tests and frequency sweep tests. Tao et al. [31] used temperature sweep tests to analyze the effects of steel slag filler on the rheological properties of asphalt mortar. Xing et al. [32] employed temperature sweep tests to study the influence of four different fibers on the rheological properties of asphalt mastics. Xu et al. [33] adopted frequency sweep tests to study high-temperature rheological properties of crumb rubber modified asphalt binders. Yan et al. [34] investigated the high-temperature rheological properties of amorphous poly alpha olefin (APAO) and ethylene-vinyl acetate copolymer (EVA) compound-modified asphalt via temperature sweep tests and frequency sweep tests. The linear amplitude sweep (LAS) test is usually used to analyze the fatigue properties of asphalt. Jafarian et al. [35] studied the fatigue properties of bitumen emulsion-cement mastics using LAS tests. Behl et al. [36] performed LAS tests to analyze the characteristic of fatigue resistance in a warm mix binder. Zhang et al. [37] investigated the fatigue resistance of aged asphalt binders based on LAS tests. Notani et al. [38] applied LAS tests to evaluate the fatigue resistance of toner-modified asphalt binders. Cao et al. [39] analyzed the fatigue characteristics of asphalt binders using LAS tests. Wang et al. [40] refined the calculation method for fatigue failure criterion of asphalt binders based on LAS tests. In summary, the DSR test is excellent for analyzing the aging of asphalt binders and asphalt mastics.

This paper focuses on the electromagnetic characteristics of limestone filler and magnetite filler and the aging behavior of magnetite filler asphalt mastics (MFAMs) under

microwave heating. A vector network analyzer was used to measure the electromagnetic characteristics of limestone filler and magnetite filler. The MFAMs were prepared and heated by microwaves. DSR tests were conducted to evaluate and analyze the high temperature rheological and fatigue properties of MFAMs after microwave heating. This study provides a reference for the application of microwave heating technology in asphalt pavement.

## 2. Materials and Methods

### 2.1. Materials

In this study, a type of 70# asphalt was selected for experimental research. Its technical indicators are listed in Table 1.

**Table 1.** Technical indicators of the asphalt.

| Technical Indicators | Unit | Measured Value | Technical Requirement |
|---|---|---|---|
| Penetration (25 °C) | 0.1 mm | 64 | 60–80 |
| Penetration index (PI) | / | −0.32 | −1.5 to +1.0 |
| Ductility (15 °C) | cm | 123 | ≥100 |
| Softening point | °C | 48 | ≥46 |
| Flash point | °C | 282 | ≥260 |
| Density (15 °C) | g·cm$^{-3}$ | 1.037 | / |

Magnetite powder was selected as a substitute for limestone filler in asphalt mastic. Their technical indicators are respectively presented in Tables 2 and 3.

**Table 2.** Technical indicators of magnetite powder.

| Technical Indicators | Unit | Measured Value |
|---|---|---|
| Apparent density | g·cm$^{-3}$ | 5.18 |
| Moisture content | % | 0.01 |
| Size range (<0.075 mm) | % | 99.9 |

**Table 3.** Technical indicators of limestone powder.

| Technical Indicators | | Unit | Measured Value | Technical Requirement |
|---|---|---|---|---|
| Apparent density | | g·cm$^{-3}$ | 2.8 | ≥2.5 |
| Moisture content | | % | 0.33 | ≤1 |
| | <0.6 mm | % | 100 | 100 |
| Size range | <0.15 mm | % | 96.8 | 90–100 |
| | <0.075 mm | % | 87.3 | 75–100 |

### 2.2. Preparation of Magnetite Filler Asphalt Mastic (MFAM)

The preparation steps of MFAM are as follows: (1) determine the mass ratio of magnetite powder to asphalt binder by replacing limestone powder with equal volume magnetite powder according to a mass ratio of 1:1 of limestone powder to asphalt binder; (2) heat asphalt binder to 160 °C using an oven, after which stir it with a high-speed mixing device; and (3) add magnetite powder into the asphalt several times during mixing and stirring until the asphalt mastic is uniformly mixed. Figure 1 depicts the mixing process of MFAM.

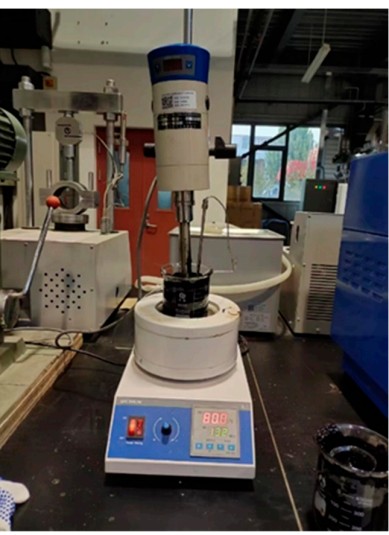

**Figure 1.** Mixing of asphalt mastics.

### 2.3. Test Methods

#### 2.3.1. Electromagnetic Parameters Measurement

The electromagnetic parameters of materials mainly include the relative complex permittivity ($\varepsilon_r$) and relative complex permeability ($\mu_r$), which are defined as Equation (1) and Equation (2), respectively.

$$\varepsilon_r = \varepsilon' - j\varepsilon'' \tag{1}$$

$$\mu_r = \varepsilon' - j\varepsilon'' \tag{2}$$

where $\varepsilon'$ and $\mu'$ are the real part of $\varepsilon_r$ and $\mu_r$ respectively, reflecting the energy storage capacity of materials; $\varepsilon''$ and $\mu''$ are the imaginary part of $\varepsilon_r$ and $\mu_r$, respectively, representing the loss ability of materials; and $j$ is imaginary unit.

The loss angle tangent value ($\tan\delta$) reflects microwave absorption ability, which can be calculated by Equations (3)–(5). The larger the $\tan\delta$, the better the microwave absorption ability.

$$\tan\delta_E = \frac{\varepsilon''}{\varepsilon'} \tag{3}$$

$$\tan\delta_M = \frac{\mu''}{\mu'} \tag{4}$$

$$\tan\delta = \tan\delta_E + \tan\delta_M \tag{5}$$

where $\tan\delta_E$ is permittivity loss angle tangent and $\tan\delta_M$ is permeability loss angle tangent.

The N5234B type vector network analyzer produced by Agilent Technologies was employed to measure complex permittivity and complex permeability of limestone filler and magnetite filler through the coaxial method in the frequency range of 2 GHz to 18 GHz. The inner diameter, external diameter, and height of coaxial ring are 3.04 mm, 7 mm, and 2 mm, respectively.

#### 2.3.2. Microwave Heating Test

The FCMCR-3C type microwave reactor with a frequency of 2.45 GHz (as shown in Figure 2) was used to heat asphalt mastics under the constant temperature mode. Microwave heating temperature was set to 165 °C. Microwave heating time for MFAM was respectively 1 h, 2 h, 3 h, and 4 h. For convenience, the asphalt mastics with the heating times of 1 h, 2 h, 3 h, and 4 h were denoted as AM-1, AM-2, AM-3, and AM-4, respec-

tively. Moreover, the properties of asphalt mastics before microwave heating (AM-0) were also analyzed.

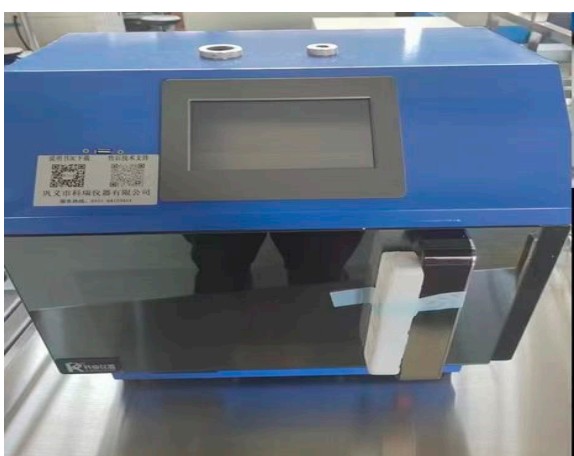

**Figure 2.** FCMCR-3C type microwave reactor.

### 2.3.3. High Temperature Rheological and Fatigue Properties Tests

The Smart Pave 102 type DSR produced by Anton Paar Company of Austria (as shown in Figure 3) was adopted in this study. The high temperature rheological properties of MFAM after microwave heating were investigated via temperature sweep tests and frequency sweep tests. The parallel-plate with a diameter of 25 mm and a gap of 1 mm was selected in the tests. The temperature sweep tests were conducted between 46 °C and 82 °C at the constant stress of 0.12 Pa. The frequency sweep tests were performed from 0.1 Hz to 10 Hz at 40 °C, 50 °C, 60 °C, and 70 °C, respectively.

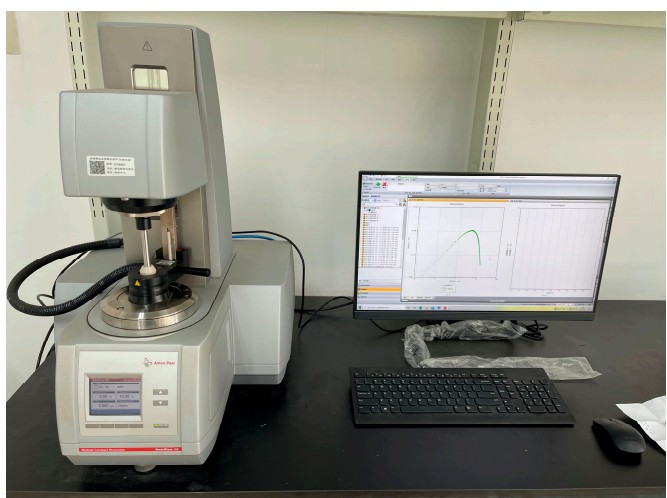

**Figure 3.** Smart Pave 102 type DSR.

The complex shear modulus (G*), phase angle (δ), and rutting factor (G* × (sin δ)$^{-1}$) based on temperature sweep tests and frequency sweep tests are three important parameters quantifying rheological properties of materials. G* reflects the ability to resist deformation of asphalt materials. The larger the G*, the better the deformation resistance of the material. δ is an index assessing the components of viscosity and elasticity of asphalt materials. The smaller the δ is, the more elasticity and the less viscosity the material has. G* × (sin δ)$^{-1}$ is an indicator reflecting the resistance to rutting of asphalt materials. The larger the G* × (sin δ)$^{-1}$, the better resistance of the material.

The fatigue properties of MFAM after microwave heating were evaluated via LAS tests. The parallel-plate with a diameter of 8 mm and a gap of 2 mm was selected in the tests. The test included two steps: (1) the frequency sweep performed from 0.2 Hz to 30 Hz at a strain of 0.1% and a temperature of 20 °C, and (2) the strain amplitude sweep conducted from 0.1% to 30% at a frequency of 10 Hz and a loading time of 10 s under each strain. According to LAS test results, non-destructive parameter $\alpha$ can be calculated by Equations (6) and (7), and cumulative damage $D(t)$ can be calculated by Equation (8).

$$\log G'(\omega) = m(\log \omega) + b \tag{6}$$

$$\alpha = 1 + \frac{1}{m} \tag{7}$$

$$D(t) \cong \sum_{i=1}^{N} [\pi I_D \gamma_0^2 (|G^*| \sin \delta_{i-1} - G^* |\sin \delta_i|)]^{\frac{\alpha}{1+\alpha}} (t_i - t_{i-1})^{\frac{1}{1+\alpha}} \tag{8}$$

where $G'(\omega)$ is storage modulus, MPa; $\omega$ is frequency, Hz; $m$ is fitting parameter; $I_D$ is complex shear modulus at a strain level of 1%, MPa; $\gamma_0$ is strain level, %; $|G^*|$ is complex shear modulus, MPa; and $t$ is test time, s.

According to viscoelastic continuum damage (VECD) theory, the relationship between $|G^*| \sin \delta(\omega)$ and $D(t)$ can be described by Equation (9).

$$\left| G^* \right| \sin \delta = C(t) = C_0 - C_1(D)^{C_2} \tag{9}$$

where $C_0$ equals 1, and $C_1$ and $C_2$ are fitting parameters.

Fatigue failure criterion $D_f$ is defined as Equation (10). Fatigue life $N_f$ can be calculated by Equations (10)–(14).

$$D_f = 0.35 \left(\frac{C_0}{C_1}\right)^{\frac{1}{C_2}} \tag{10}$$

$$A_{35} = \frac{f(D_f)^k}{k(\pi I_D C_1 C_2)^{\alpha}} \tag{11}$$

$$k = 1 + (1 - C_2)\alpha \tag{12}$$

$$B = -2\alpha \tag{13}$$

$$N_f = A_{35}(\gamma_{\max})^B \tag{14}$$

where $f$ is loading frequency (10 Hz) and $\gamma_{\max}$ is the estimated maximum strain of pavement structure, %.

## 3. Results and Discussion

### 3.1. Microwave-Absorbing Ability of Limestone and Magnetite

Figure 4 depicts the electromagnetic parameters of limestone filler and magnetite filler at different frequencies. As can be seen from Figure 4a,b, the values of $\varepsilon'$ and $\varepsilon''$ for magnetite filler are between 4.72 and 5.39 and 0.07 and 0.51, respectively, and those for limestone filler are between 3.42 and 3.67 and 0 and 0.16, respectively, indicating that the magnetite filler has a better storage ability of electric field energy and higher dielectric loss than the limestone filler. With the increase in frequency, $\varepsilon'$ changes slightly, while $\varepsilon''$ changes largely. As can be seen from Figure 4c, the value of $\mu'$ for the magnetite filler is larger than that for the limestone filler when the frequency is smaller than 4.8 GHz, indicating that the magnetite filler has better magnetic properties than the limestone filler

at low frequencies, while the opposite is true when the frequency is greater than 4.8 GHz. As can be seen from Figure 4d, the value of $\mu''$ for the magnetite filler is between 0 and 0.29 and larger than that of the limestone filler in general, indicating that the magnetite filler has a higher magnetic loss than limestone filler. Moreover, with the increase in frequency, both $\mu'$ and $\mu''$ of the magnetite filler show a wavelike decrease.

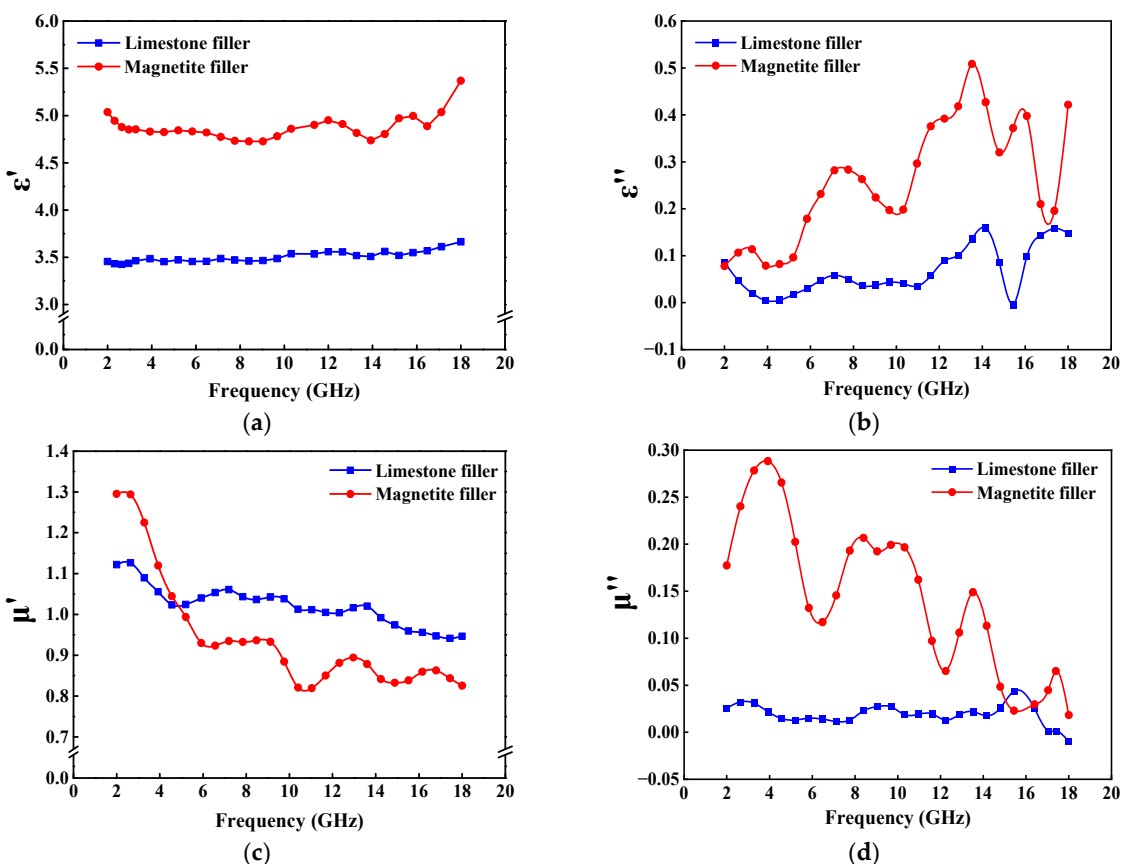

**Figure 4.** Change of electromagnetic parameters of limestone filler and magnetite filler with frequency: (**a**) $\varepsilon'$; (**b**) $\varepsilon''$; (**c**) $\mu'$; and (**d**) $\mu''$.

Figure 5 describes the permittivity loss angle tangent and the permeability loss angle tangent of the limestone filler and the magnetite filler at different frequencies. It can be seen that the $\tan\delta_E$ of the magnetite filler is between 0.01 and 0.11. It has the smallest value at a frequency of 4 GHz and the largest value at a frequency of 14 GHz. The $\tan\delta_E$ of the limestone filler is between 0 and 0.05. The $\tan\delta_M$ of the magnetite filler is between 0 and 0.26 and that of the limestone filler is between 0 and 0.05. In general, the magnetite filler has a larger $\tan\delta_E$ and $\tan\delta_M$ than the limestone filler. Furthermore, the $\tan\delta_E$ and $\tan\delta_M$ of the magnetite filler change significantly with frequency, while those of the limestone filler change slightly.

Figure 6 shows the $\tan\delta$ of the limestone filler and the magnetite filler at different frequencies. As can be seen in the figure, the $\tan\delta$ of the magnetite filler is between 0.05 and 0.28, and that of the limestone filler is between 0.01 and 0.07. The value of $\tan\delta$ for the magnetite filler is the largest at a frequency of 4 GHz. The magnetite filler has a larger $\tan\delta$ than the limestone filler, indicating that the magnetite filler has a better microwave absorption ability. The $\tan\delta$ of the magnetite filler changes significantly with frequency, while that of the limestone filler changes slightly.

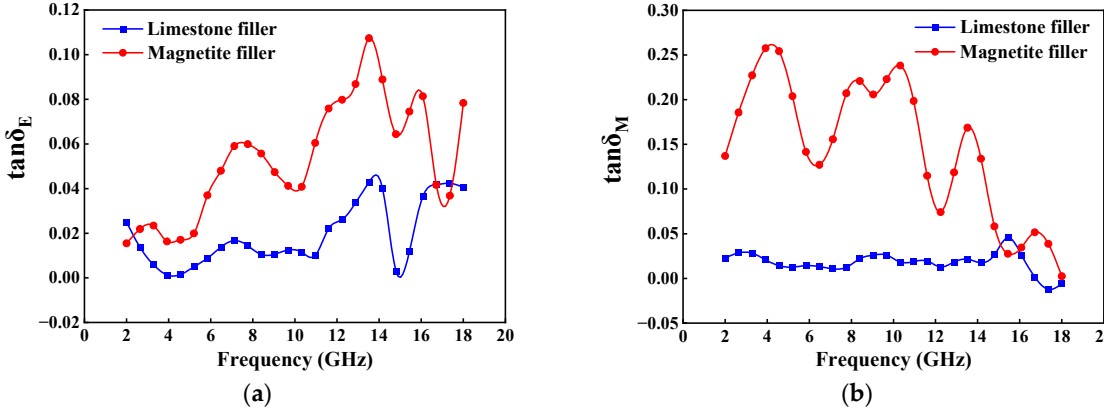

**Figure 5.** Change of tan$\delta_E$ and tan$\delta_M$ of limestone filler and magnetite filler with frequency: (**a**) tan$\delta_E$ and (**b**) tan$\delta_M$.

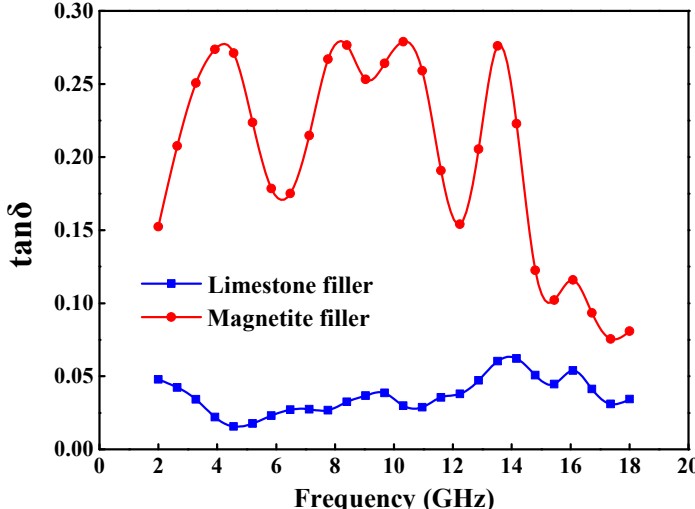

**Figure 6.** Change of tan$\delta$ of limestone filler and magnetite filler with frequency.

In summary, both the microwave-absorbing ability and microwave sensitivity of magnetite are significantly better than those of limestone. Therefore, using magnetite powder as a substitute for limestone filler in asphalt mastic can substantially enhance microwave sensitivity and improve microwave heating efficiency.

### 3.2. Temperature Sweep Test Results

Figure 7 presents the G* and δ of the MFAMs after microwave heating at different temperatures. As can be seen from Figure 7a, G* rapidly decreases with the increase in temperature, indicating that the deformation resistance of the MFAMs drops sharply as temperature increases. The reason for this is that rising temperature accelerates the fluidity of the asphalt binder. The G* of asphalt mastics containing common fillers also has the same varying tendency with temperature [32], which illustrates that magnetite powder does not change the trend. At 46 °C–52 °C, the longer the microwave heating time, the larger the G* of the MFAMs. Moreover, the longer the microwave heating time, the faster the G* of the MFAMs decreased with temperature rising. This means that microwave heating has a significant influence on the G* of the MFAMs. However, the influence diminishes gradually with the increase in temperature.

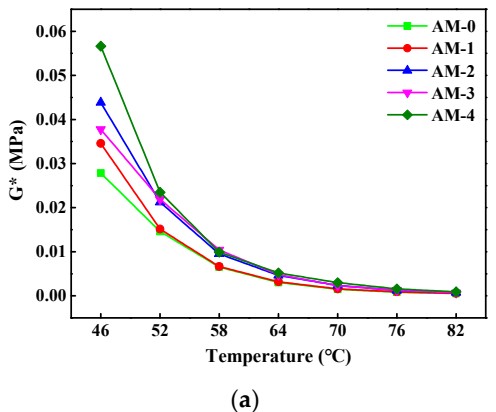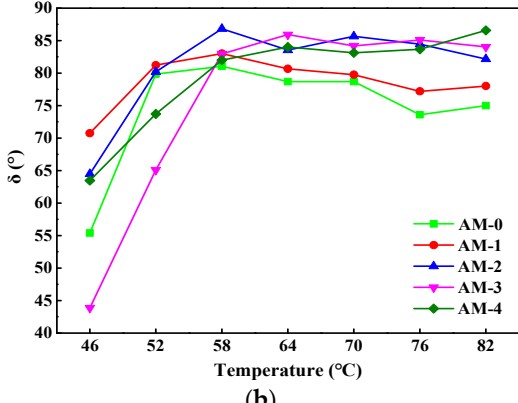

(**a**)　　　　　　　　　　　　　　　　　　　　　　　(**b**)

**Figure 7.** Complex shear modulus and phase angle of MFAMs at different temperatures: (**a**) complex shear modulus and (**b**) phase angle.

As can be seen from Figure 7b, the δ of AM-0, AM-1 and AM-2 gradually increases at 46 °C–58 °C, and that of AM-3 and AM-4 gradually increases at 46 °C–64 °C, and that of all MFAMs keep relatively steady at 58 °C–82 °C. The results mentioned above indicate that the elasticity of the MFAMs decreases, and the viscosity increases with the increase in temperature. The composition of elasticity and viscosity remains basically unchanged when temperature reaches a certain threshold. The δ of AM-2, AM-3, and AM-4 is almost the same and larger than that of AM-0 and AM-1 at 64 °C–82 °C, which indicates that microwave heating for over two hours has significant effects on the δ of MFAMs.

Figure 8 depicts the $G^* \times (\sin \delta)^{-1}$ of the MFAMs after microwave heating at different temperatures. It can be seen that $G^* \times (\sin \delta)^{-1}$ rapidly decreases with the increase in temperature, indicating that the anti-rutting performance of MFAMs drop sharply with a temperature increase. At 46 °C–52 °C, the MFAMs with longer microwave heating time have a larger $G^* \times (\sin \delta)^{-1}$. The reason for this is that microwave heating hardens the MFAMs. Moreover, the longer the microwave heating time, the faster the $G^* \times (\sin \delta)^{-1}$ decreased with temperature rising. The effect of microwave heating on $G^* \times (\sin \delta)^{-1}$ gradually decreased with the increase in temperature.

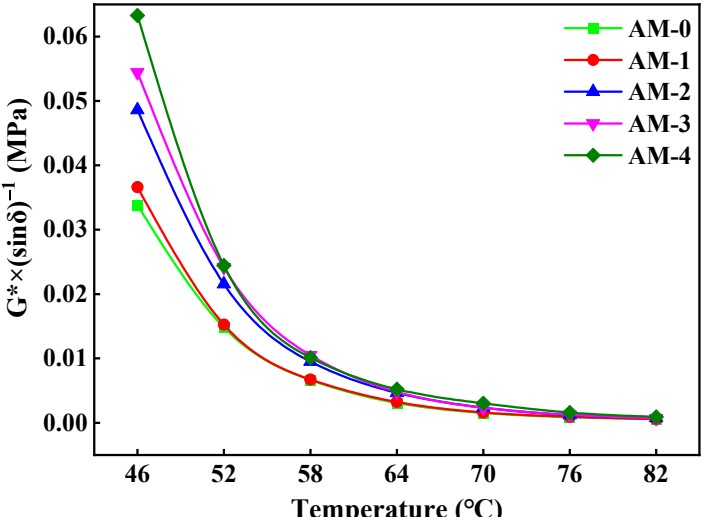

**Figure 8.** Rutting factor of MFAMs at different temperatures.

### 3.3. Frequency Sweep Test Results

Figure 9 describes the G* of MFAMs after microwave heating time at different frequencies. As can be seen from the figure, the G* of each MFAM is relatively small at

low frequencies and increases with an increase in frequency. This means that the MFAM provides greater rigidity with the increase in vehicle speed. And, it is not compressed instantaneously during the loading process, and does not recover instantaneously during the unloading process. The higher the frequency, the more significant the effect of microwave heating time on G*.

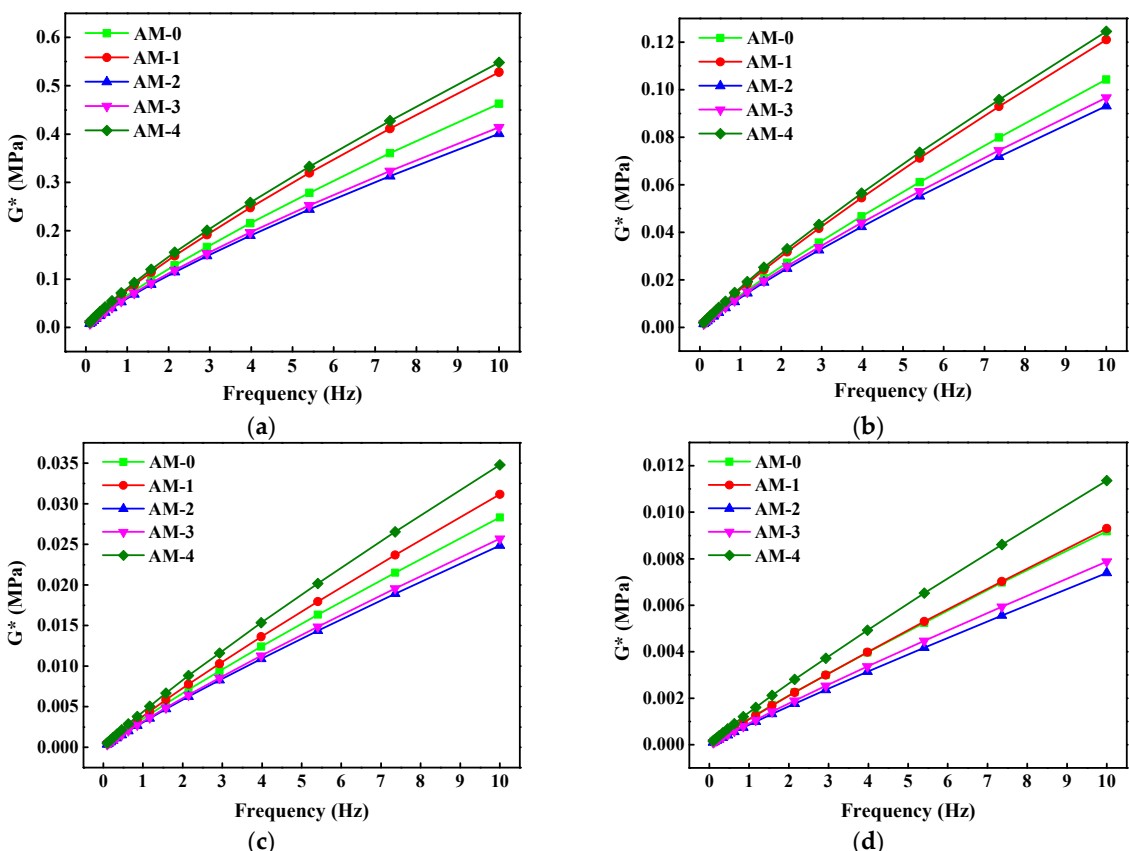

**Figure 9.** Complex shear modulus of MFAMs at different frequencies: (**a**) at 40 °C; (**b**) at 50 °C; (**c**) at 60 °C; and (**d**) at 70 °C.

Figure 10 depicts the δ of the MFAMs after microwave heating at different frequencies. It is shown that δ gradually decreases as the frequency increases, indicating that the MFAM shows more elasticity with an increase in vehicle speed. The higher the frequency, the more significant the effect of microwave heating time on δ.

Figure 11 presents the $G^* \times (\sin \delta)^{-1}$ of MFAMs after microwave heating at different frequencies. It can be seen that $G^* \times (\sin \delta)^{-1}$ increases with the increase in frequency. The $G^* \times (\sin \delta)^{-1}$ of each MFAM is relatively small at low frequencies. The higher the frequency, the more significant the effect of microwave heating time on $G^* \times (\sin \delta)^{-1}$.

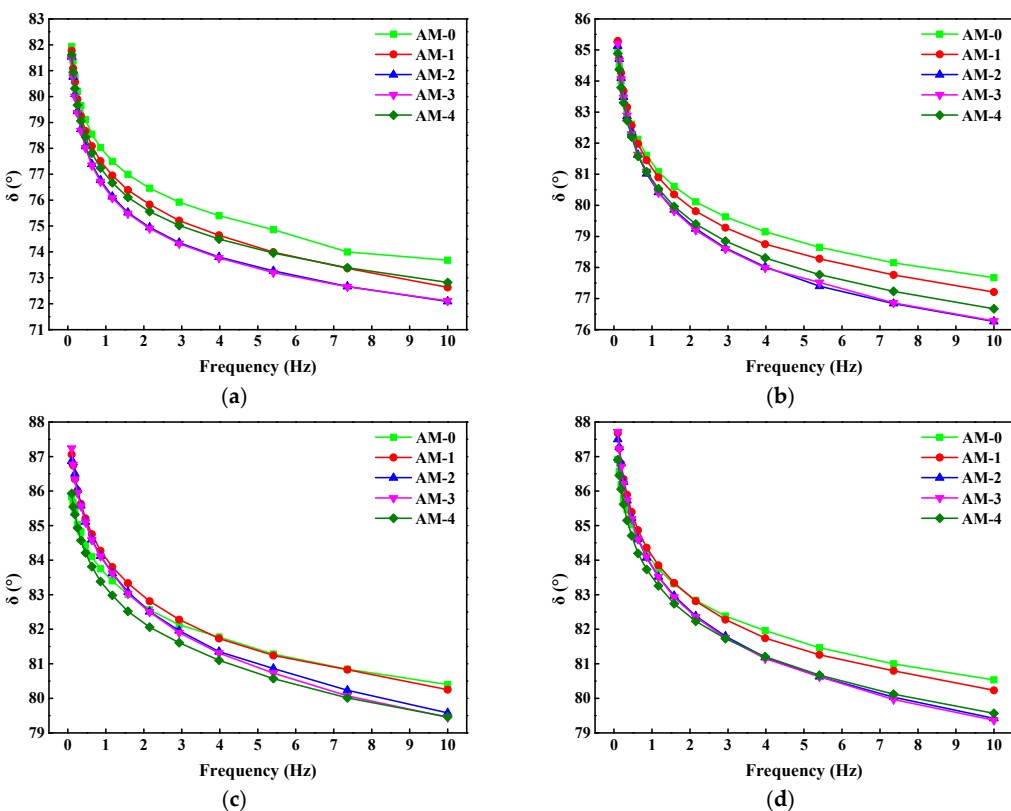

**Figure 10.** Phase angle of MFAMs at different frequencies: (**a**) at 40 °C; (**b**) at 50 °C; (**c**) at 60 °C; and (**d**) at 70 °C.

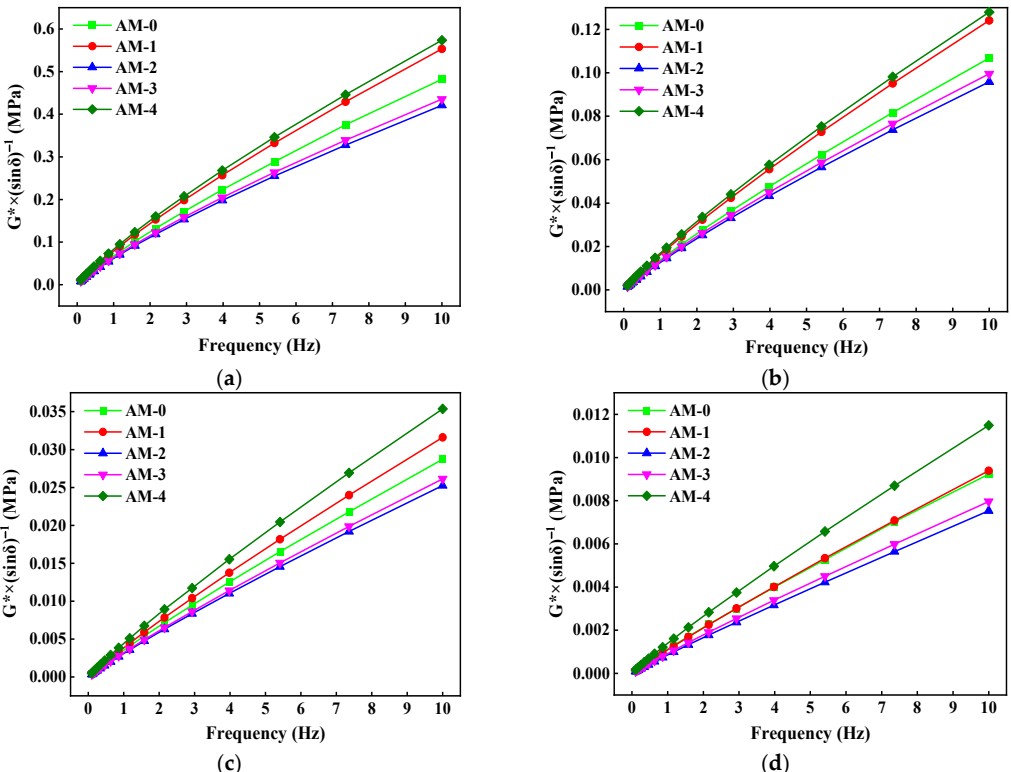

**Figure 11.** Rutting factor of MFAMs at different frequencies: (**a**) at 40 °C; (**b**) at 50 °C; (**c**) at 60 °C; and (**d**) at 70 °C.

### 3.4. Linear Amplitude Sweep Test Results

Figure 12 presents the stress–strain curves of the MFAM after microwave heating. As can be seen from the figure, the shear stress of the MFAM has a peak value. The stress–strain curve shape of the various MFAMs is basically the same. The shear stress first increases to a peak stress rapidly, and then drops to the lowest value sharply, and finally rebounds slowly. The peak shear stress of AM-0 is far larger than that of the other four MFAMs. The yield strain, the shear strain corresponding to the peak shear stress, is about 3% for AM-0 and AM-1, and about 2% for AM-2, AM-3, and AM-4. The results mentioned above indicate that microwave heating reduces the peak shear stress and yield strain of the MFAM.

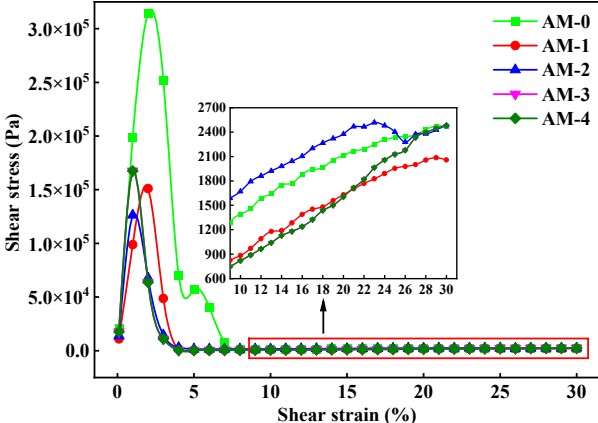

**Figure 12.** Stress–strain curves of MFAMs.

Table 4 lists the VECD model parameters of the MFAMs. As shown in the table, microwave heating promotes $C_2$ and diminishes $A_{35}$ and $C_1$, and it has little influence on $\alpha$ and B. Figure 13 presents the fatigue life of the MFAMs at the strains of 2.5% and 5%. As can be seen from the figure, the fatigue life of the MFAMs obviously decreases with an increase in strain and microwave heating time.

**Table 4.** VECD model parameters of MFAMs.

| Asphalt Mastics | $\alpha$ | $A_{35}$ | B | $C_1$ | $C_2$ |
|---|---|---|---|---|---|
| AM-0 | 2.26 | $1.48 \times 10^7$ | −4.52 | 0.2015 | 0.4666 |
| AM-1 | 2.28 | $9.89 \times 10^5$ | −4.55 | 0.1474 | 0.5254 |
| AM-2 | 2.29 | $4.01 \times 10^5$ | −4.58 | 0.1599 | 0.5413 |
| AM-3 | 2.27 | $1.22 \times 10^5$ | −4.54 | 0.0918 | 0.6418 |
| AM-4 | 2.27 | $1.02 \times 10^5$ | −4.54 | 0.0815 | 0.6922 |

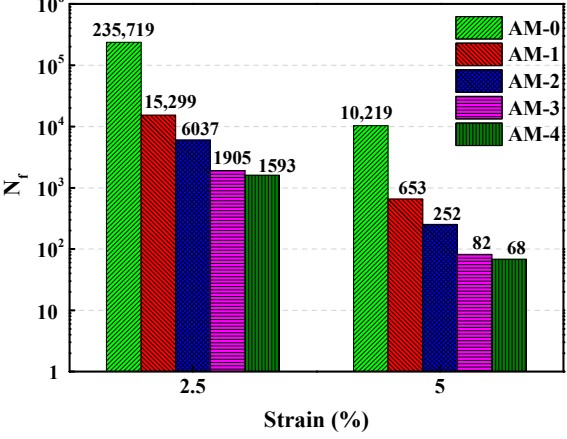

**Figure 13.** Fatigue life of MFAMs at 2.5% strain and 5% strain.

## 4. Conclusions

In this study, the electromagnetic parameters of limestone and magnetite were measured. Temperature sweep tests, frequency sweep tests, and LAS tests were performed to evaluate the high temperature rheological and fatigue properties of MFAMs. The following conclusions can be drawn:

(1) Magnetite has a better electric field energy storage ability, a higher dielectric loss and magnetic loss, and a better microwave heating efficiency than limestone. And, it has better magnetic properties than limestone at a low frequency, while the opposite is true at a high frequency. The $\tan\delta$ of magnetite changes significantly with frequency, and that of limestone changes slightly.

(2) With the increase in temperature, the deformation resistance and anti-rutting performance of the MFAM drops sharply, its elasticity decreases, and its viscosity increases. Its composition of elasticity and viscosity remains relatively steady when the temperature reaches a certain threshold. The longer the microwave heating time, the faster the high temperature rheological properties of the MFAM decreased with rising temperature.

(3) Microwave heating reduces the peak shear stress and yield strain of the MFAM. The fatigue life significantly decreases with the increase in strain and microwave heating time.

Microwave heating technology has good prospects in promoting asphalt pavement self-healing and de-icing. However, microwave heating can also result in asphalt aging. It is suggested to completely evaluate the aging of asphalt pavements under cycling microwave heating and develop a suitable anti-aging agent to reduce the aging degree of asphalt in future research.

## 5. Scope for the Future Work

(1) XRD analysis, XPS studies, FTIR analysis, and morphological analysis for magnetite powder should be conducted to evaluate the physical characterization and reveal enhancing mechanisms of microwave sensitivity.

(2) The performance between MFAMs subjected to microwave heating and conventional asphalt materials should be compared. Long-term performance, cost-effectiveness, and environmental impact when using microwave heating should be calculated.

(3) The effect of magnetite powder proportion on the performance of asphalt mastics after microwave heating should be explored.

**Author Contributions:** Conceptualization, Y.M.; data curation, L.Z.; funding acquisition, Y.M.; methodology, W.Y., L.Z. and Y.M.; supervision, Y.M. and S.W.; validation, Z.G. and S.W.; writing—original draft, W.Y. and S.W.; writing—review and editing, W.Y., L.Z. and Z.G. All authors have read and agreed to the published version of the manuscript.

**Funding:** This work was funded by the National Key R&D Program of China (grant number: 2019YFE0117600).

**Institutional Review Board Statement:** Not applicable.

**Informed Consent Statement:** Not applicable.

**Data Availability Statement:** Not applicable.

**Conflicts of Interest:** The authors declare no conflict of interest.

**Abbreviations**

| | |
|---|---|
| LAS | linear amplitude sweep |
| VECD | viscoelastic continuum damage |
| APSH | asphalt pavement self-healing |
| APAO | amorphous poly alpha olefin |
| EVA | ethylene-vinyl acetate copolymer |
| MFAM | magnetite filler asphalt mastic |
| AM-0 | asphalt mastic without microwave heating |
| AM-1 | asphalt mastic with the heating time of 1h |
| AM-2 | asphalt mastic with the heating time of 2h |
| AM-3 | asphalt mastic with the heating time of 3h |
| AM-4 | asphalt mastic with the heating time of 4h |

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
