# Peer review of "Microwave Sensitivity Enhanced Asphalt Mastic with Magnetite Powder and Its Performance after Microwave Heating"

_applsci, doi:10.3390/app13148276_

Round 1
Reviewer 1 Report
The manuscript entitled "Microwave Sensitivity Enhanced Asphalt Mastic with Magnetite Powder and Its Performance After Microwave Heating" describes about the various physical and mechanical properties of magnetite based composites. In terms of the topic and the concept, the results and the way the authors interpretated is very good. I see that there are no significant comments to be made on the work. As far as the theme is concerned, they stick to that one and very well delivered the point. Any additional comments/suggestions may distract from the core theme.
However, I want to suggest the authors to add some physical characterization like the powdered XRD analysis, XPS studies, FTIR analysis, and most importantly the morphological studies by making use of HRTEM/FESEM analysis. Without these analysis, it is a bit difficult to come to a conclusion. We need to confirm whatever the results that they presented are exactly the same material and in the same phase. Without such physicochemical studies, it may not be a good idea to generate a conclusion. Therefore, I suggest to include these analysis when they revise the work and I feel they are most important for this particular work to have a cumulative idea.
Reviewer 2 Report
This manuscript explores the “Microwave Sensitivity Enhanced Asphalt Mastic with Magnetite Powder and Its Performance After Microwave Heating”. The manuscript is elaborately described and contextualized with the help of previous and present theoretical background. All the references cited are relevant to this area of research. The methods/analytical study are clearly stated. The result and discussion section are clearly presented. The manuscript needs the following modifications before the acceptance.
1. Abstract – include research recommendation, Significance, and impact of the work.
2. Key words: arrange the keywords in alphabetical order.
3. State the novelty of your work.
4. Fig.6. Present Y axis value in MPa and do the same throughout the manuscript.
5. Include the experimental photos.
6. Conclusion: Include your research recommendations
7. Include a section “Scope for the future work”
8. Include a section “List of abbreviations”
Minor editing of English language required
Reviewer 3 Report
Please read the attachment. Thank you.

Minor changes are needed.
Reviewer 4 Report
The manuscript presents an interesting topic for pavement engerineers. The reviewer has some sugestions to improve the quality of the manuscript
The abstract should be improved by introducing the main findings of the study
The research objectives and research gab should be highlighted.
The results may be different if the percent of asphalt to filler changed. Please clarify the influence of changing this ratio on the results.
Using the microwave as heating tool on the mass production scale has a negative impact on the environment related to the radiation.
A limitations section should be added to the manuscript.
The English language should be improved and edited by a native speaker.
Round 2
Reviewer 1 Report
The manuscript can now be accepted for the final publication.
Reviewer 4 Report
The authors addressed the reviewer comments